# Marginal Adaptability of Harvard MTA and Biodentine Used as Root-End Filling Material: A Comparative SEM Study

**DOI:** 10.3390/ma18194598

**Published:** 2025-10-03

**Authors:** Yaneta Kouzmanova, Ivanka Dimitrova

**Affiliations:** Department of Conservative Dentistry, Faculty of Dentistry, Medical University, 1431 Sofia, Bulgaria; vanja_ves@abv.bg

**Keywords:** calcium silicate cements, Harvard MTA, marginal adaptation, scanning electron microscopy

## Abstract

The proper selection of bioactive root-end material is one of the main prognostic factors for the successful healing outcome of apical microsurgery (AMS). The aim of the present in vitro study was to evaluate and compare the marginal adaptability of a novel calcium silicate cement (CSC), Harvard MTA Universal, and Biodentine used as root-end filling materials. The endodontic treatment of 20 extracted human maxillary central incisors was performed. The apicoectomy was simulated, and root-end cavities were prepared ultrasonically using universal retrotips. Teeth were randomly assigned into two equal groups (n = 10) according to the retrofilling cement used: Group 1—Harvard MTA Universal and Group 2—Biodentine. The specimens were stored in relative humidity for 48 h and sectioned longitudinally. The data were processed and analyzed statistically. Harvard MTA exhibited a significantly lower mean gap width (1.16 ± 0.37 µm) than Biodentine (2.48 ± 0.38 µm) (*p* < 0.05), indicating a more intimate interfacial adaptation. Additionally, the phenomenon of material penetration into the dentinal tubules was observed only in the Harvard MTA group. Within the limitations of this in vitro study, Harvard MTA Universal demonstrated better interfacial properties than Biodentine when applied as a root-end filling material. This novel biomaterial could be regarded as a promising alternative for earlier calcium silicate cements in the context of AMS goals. **Clinical relevance:** The quality of marginal adaptation is a determinative feature for the clinical performance of CSCs and the long-term prognosis of AMS.

## 1. Introduction

Apical microsurgery (AMS) may be the only treatment modality to preserve endodontically compromised natural teeth. The ultimate goal of the modern surgical endodontic treatment is to hermetically seal the root canal system in order to prevent microleakage and the recontamination of the periapical area [1,2]. One of the prognostic factors determining healing outcome is the proper selection of a bioactive root-end material with appropriate characteristics [1,3]. Recently, mineral trioxide aggregate (MTA) and its derivatives, termed calcium silicate cements (CSCs), have become the material of choice for root-end filling due to their physical and biological advantages over older materials [4,5,6]. One of the determinative factors for the clinical performance of this class of biomaterials and long-term prognosis of AMS is their good marginal (interfacial) adaptability to the dentin. This peculiar feature denotes the nature of interfacial interrelationship between the cement and dentin walls [6,7,8].

Biodentine, which was introduced in 2009, is an outstanding representative of the third generation of CSCs, also referred to as Bioceramics. It was developed as a dentine substitute with the intention of transcending previous CSCs and potentially replacing them for certain endodontic purposes. Biodentine is considered to have superior properties to earlier MTA products due to its distinct chemical composition. A plethora of studies suggests its better sealing capacity and clinical effectiveness, surpassing conventional CSCs in many applications, including AMS [9,10,11,12]. Currently, it is regarded as a new gold standard for the biomedical research of the bioceramics group, on the same level as ProRoot MTA, which was the first MTA brand introduced.

A number of new CSCs have been introduced into the dental market during the last decade, many of them with promising results in AMS [13,14,15]. Harvard MTA Universal is an innovative general-purpose bioactive calcium silicate-based cement. A thorough literature search revealed rather limited evidence on its properties, mainly in terms of chemical content, solubility, dislodgement resistance, and cytotoxicity [16,17,18,19,20]. However, the sealing capacity of this novel MTA brand has not yet received sufficient attention regarding its relative significance to existing CSCs in terms of interfacial adaptability [21,22].

Our previous pilot micro-CT study, confined to Harvard MTA only, addressed the volumetric estimation of its interfacial and internal porosity [23]. Therefore, the aim of the present study was to evaluate and compare the marginal adaptability of Harvard MTA Universal and Biodentine, used as root-end filling materials, through scanning electron microscopy (SEM). Most CSCs are based predominantly on tricalcium silicate (TCS) [8,24]. Accordingly, the null hypothesis stated that the linear gap width produced by the two hydraulic biomaterials under test did not differ significantly, based on the similarity in their chemical compositions.

## 2. Materials and Methods

This in vitro study was designed and conducted as a double-blind randomized controlled trial, according to modified CONSORT guidelines for in vitro studies. The study design was adapted to align with the most commonly used experimental conditions found in previous SEM-based studies and our earlier research on this topic. The study protocol is presented schematically in a flowchart (Figure 1). It was first presented in our previous works [23,25], and its description partly reproduces their wording. However, certain enhancements were also made to this study’s methodology and equipment.

### 2.1. Sample Construction

A total of 20 freshly extracted human maxillary central incisors were used in this study, obtained from the Department of Oral and Maxillofacial Surgery at the Faculty of Dental Medicine, Sofia. The assessment and the preselection of suitable teeth were performed using a clinical and radiologic method. The inclusion criteria for the teeth were as follows: (1) a single straight root with a facial root length of 14 mm; (2) a patent single canal; (3) a mature root with a closed apex; and (4) an intact crown and a pulp chamber. Exclusion criteria: (1) curved or fractured root or open apex; (2) calcified root canal; (3) prior root canal treatment; (4) any type of root resorption; or (5) visible cracks on examination. After extraction, the teeth were fixed in 10% buffered formalin for 2 weeks. Afterwards, they were cleaned manually and ultrasonically, rinsed, and stored in a 0.2% thymol solution before the next steps.

### 2.2. Endodontic Preparation

A clinically oriented approach was applied by preserving the tooth crown’s integrity. Nevertheless, specimen standardization was achieved through morphometric preselection. The endodontic preparation of the teeth was performed using a rotary NiTi system, ProTaper Next (up to X3/0.07 file), and corresponding calibrated F3/0.06 gutta-percha points (Dentsply Sirona, Ballaigues, Switzerland) and root canal sealer AH Plus (Dentsply DeTrey GmbH, Konstanz, Germany). Postoperative radiographs were taken to confirm the fillings’ homogeneity. After definitive occlusal restorations with 3M Filtek™ Z250 Universal Restorative (3M ESPE Dental Products, St. Paul, MN, USA), the teeth were stored in 0.9% saline (B. Braun Melsungen AG, Melsungen, Germany) for 2 days.

### 2.3. Root-End Cavity Preparation and Root-End Filling

An apical resection was simulated by cutting the apical root portion at 90° 3 mm short of the root tip using a high-speed Zekrya surgical carbide bur FG (Dentsply Sirona Group, Ballaigues, Switzerland) under continuous saline cooling. Standardized retrograde cavities (3 mm depth, d = 1.2 mm) were prepared coaxially into the canal space with a diamond-coated micro retrotip P14D (Acteon Satelec, Mérignac, France) using a piezoelectric ultrasonic device, Suprasson P5 Booster (Acteon Satelec, Mérignac, France).

At this point, teeth were randomly assigned into two equal groups (n = 10) using a simple randomization method (coin flip). In each group, the root-end cavities were filled with their respective CSC as follows: Group 1—**Harvard MTA Universal HandMix** (Harvard Dental International GmbH, Hoppegarten, Germany); Group 2—**Biodentine** (Septodont, Saint-Maur-des-Fossés Cedex, France). In accordance with the manufacturers’ instructions, Harvard MTA (henceforward used in this text for brevity) was prepared manually with a powder/liquid ratio of 2.6/1.0 by weight, while Biodentine was prepared in a capsule mixer (Dubai Medical Equipments, Dubai, United Arab Emirates). The cements were inserted into the root-end cavity incrementally using a MAP One syringe (Micro-apical Placement System, Produits Dentaires SA, Vevey, Switzerland). All the surgical procedures were completed under a dental microscope, KAPS SOM^®^ 62 1100 (Karl Kaps GmbH & Co. KG, Asslar/Wetzlar, Germany).

The quality of the retro-fillings throughout the sample was verified radiographically in two projections and assessed by two previously calibrated examiners (Cohen’s Kappa, k = 0.92). Afterwards, the specimens were incubated in 100% relative humidity at 37 °C for 48 h. In the next step, the teeth were sectioned longitudinally in the mesiodistal direction through the center of the material using a slow-speed diamond disk (PHM, Plovdiv, Bulgaria) under copious amounts of water spray.

### 2.4. Marginal Adaptation Assessment (Scanning Electron Microscopy Analysis)

Prior to scanning, all specimens were mounted on aluminum stubs and sputter-coated with a gold layer (10 nm thickness), using a JEOL JFC-1200 Fine Coater (JEOL Ltd., Tokyo, Japan). Specimens were examined under a TM400 high-vacuum scanning electron microscope (SEM) (Hitachi, Tokyo, Japan), at an accelerating voltage of 15 kV. Different magnifications were used—from ×30 to ×2000—to assess the marginal adaptation of the tested biomaterials. The images obtained at magnification ×30 served for the determination of four symmetrical points along the interface at the 1 mm and 2 mm levels from the resected apex. Further, these magnified areas (×1000) were used as a base for the complex estimation of the dentin-to-material interface characteristics. Overall, 16 measurements per specimen were made (Figure 2), i.e., 160 per group.

A built-in function in the TM400 software was used to measure the marginal gaps’ width (a cement-to-dentin wall distance) throughout the sample. Quantification was performed by an experienced operator who was blinded to the nature and type of the materials. Two specimens per group were used for training on the measuring procedure, with provisional measurements taken by the SEM technician alongside the researchers to set proper criteria. The assessment of these first 64 micrographs facilitated intra-examiner calibration. Final gap quantifications were carried out in real time, directly on screen, with recorded values captured in SEM images. The measurement’s accuracy was independently rated by two examiners, with images classified as “acceptable” or “unacceptable.” Disagreements led to the replacement of questionable images with additional ones. Cohen’s Kappa coefficient showed almost perfect inter-rater agreement (k = 0.85). Lastly, the mean gap width for each specimen was computed.

### 2.5. Statistical Analysis

All data were entered and processed with the statistical package IBM SPSS Statistics 25.0 (SPSS, Inc., Chicago, IL, USA), as the statistician was blinded as well. Descriptive statistics were used to assess the gap size produced by the two CSCs under evaluation. The results were also submitted to the following tests: (1) a Shapiro–Wilk test to check the normality of data distribution; (2) Student’s independent samples *t*-test to evaluate the significance of the differences between the groups; and (3) Cohen’s d statistics to calculate the magnitude of the effect size. The α-level was set at 0.05, as a *p*-value less than 0.05 (*p* < 0.05) was considered statistically significant. Additionally, Cohen’s Kappa coefficient was used for the inter-rater reliability assessment.

## 3. Results

### 3.1. Qualitative Evaluation of Material-to-Dentine Interface

The SEM examination of the longitudinal sections revealed a clear picture of the materials and the root dentin cut surfaces. Representative SEM images for the interfacial areas in both groups at magnifications of ×500 and ×1000 are presented in Figure 3. The bulk material in the Biodentine group seemed more homogeneous, and the size of the radiopacifier particles was more uniform. Conversely, the Harvard MTA structure showed a marked heterogeneity, with a small number of large polygonal crystals surrounded by an amorphous mass of smaller crystals. Some specimens in Group 1 had material levels below the dentinal wall’s cut surface, suggesting that the material’s abrasion resistance is likely lower.

Both materials appeared to be well adapted to the dentinal walls overall, with no significant voids seen in the interfacial zone, but Biodentine showed more noticeable gaps. There were no samples without a gap; nonetheless, zones with barely noticeable gaps of minor size were observed mainly with Harvard MTA. Particularly in this group, some specimens’ gaps seemed to be filled by the material or by an amorphous substance. Furthermore, an occurrence of a phenomenon of biomaterial penetration into the adjacent dentin tubules was also detected only in the MTA group (Figure 3).

### 3.2. Quantitative Assessment of Interfacial Gaps

Harvard MTA exhibited a lower mean gap width (1.16 ± 0.37 µm) compared to Biodentine (2.48 ± 0.38 µm), with a statistically significant intergroup difference (*p* < 0.001), which indicated a more intimate interfacial adaptation. Table 1 summarizes descriptive statistics parameters for both groups. Additionally, Figure 4 displays the results graphically in a bar chart. The better performance of Harvard MTA was confirmed by the Cohen’s d index (d = 3.52), indicating the very large effect size (d ≥ 1.3) of the material’s type. The boxplot diagram (Figure 5) showed a more symmetrical data distribution in the Harvard MTA group, suggesting a balanced spread of the specimens’ mean values. In addition, the 95% Confidence interval produced in the Harvard MTA group [0.89, 1.42] showed lower boundaries in comparison with the Biodentine group [2.20, 2.75].

## 4. Discussion

The present in vitro SEM study found a significantly lower interfacial gap width with Harvard MTA Universal when compared with Biodentine (1.16 μm vs. 2.48 μm, respectively; *p* < 0.05), suggesting a better root-end marginal adaptability of the former. Therefore, the null hypothesis tested was rejected. Only two previous studies on Harvard MTA were found, addressing its interfacial characteristics evaluated through an SEM. Regardless of some differences in the testing conditions, both works reported very similar mean values of the marginal gap size exhibited by Harvard MTA—7.44 and 8.53 µm, respectively [21,22]. These results are significantly higher and in contrast with our findings, which in turn are supported by our previous micro-CT volumetric examination of the interfacial and internal porosity of this novel restorative cement [23].

By contrast, Biodentine is probably the most extensively investigated biomaterial among bioceramics in the scope of apical microsurgery. Regarding its marginal adaptation, this cement has shown considerable variability in performance, with gap widths covering the spectrum from 0.46 to 22.41 μm [26,27]. Most of the previous research found significantly lower gap widths, ranging from 0.46 to 1.38 μm [10,26,27,28,29,30], and are not consistent with our results. Conversely, other studies indicated a larger gap size (3.147–22.41 μm), compared to the findings of the present study [22,27,31,32,33]. A single paper was found, by Bansal et al. [34], whose reporting results—2.44 μm—were equivalent to ours, as the latter validated our prior SEM investigation of Biodentine [25].

Furthermore, it is of interest to consider the relative sealing efficacy of Biodentine and Harvard MTA compared to other investigated CSCs in terms of gap size. According to several recent studies, Biodentine exhibited superior marginal adaptation in comparison with ProRoot MTA, MTA-Angelus, MTA Plus, and Retro-MTA [10,27,34,35]. Most studies, however, established an equivalent sealing performance of Biodentine and other CSCs—ProRoot MTA, MTA Angelus, Bioaggregate, CEM cement, and iRoot BP Plus [28,30,31,36].

In contrast to the aforementioned research, a few investigations found larger gap widths with Biodentine compared to ProRoot MTA or other new brands of CSCs [29,32,33]. The present study results are consistent with them. As for Harvard MTA, according to the limited previous studies, it has demonstrated a similar interfacial adaptability to Biodentine [22] but worse than MTA Flow [21].

As seen, there is no agreement on the marginal adaptability of Biodentine when used as a root-end filling. These inconsistencies are commonly attributed to the lack of methodology standardization. Great variability exists in the study designs in terms of specimen nature and its preparation, storage time, root sectioning plane, type of SEM used, gap measurement, etc. [28,35,36]. The impact of the sectioning plane on the accuracy of the results seems to be highly controversial, with the findings of Bolbolian et al. [27] being a very vivid example of this. Biodentine has shown the best results when assessed in a longitudinal plane (4.49 μm) but the worst in transversal sections (22.41 μm).

Reportedly, the gap width found with Biodentine in the transversal plane varied more widely—0.46-22.41 μm [10,26,28,29,30,31,32] vs 2.44-8.53 μm for the longitudinal one [22,33,34]. Our findings fall at the lower end of this spanning interval regarding other studies that have used longitudinal sectioning. As for Harvard MTA, the difference in the reported gap size values for both planes is not so pronounced—7.44 μm and 8.17 μm, respectively [21,22].

The transversal plane is most commonly preferred as a base for the gap quantification as it provides assessment for the entire circumference of the marginal gap [10,21,30,33]. On the other hand, the longitudinal plane offers insight along the complete length of the interfacial zone. According to our previous micro-CT study, Harvard MTA demonstrated the highest density predominantly in middle portion of the root-end filling, while the coronal part exhibited the lowest density [23]. These observations are in line with some earlier SEM studies, reporting a notable gap size heterogeneity at the various root levels—1 and 2 mm from the resected apex [10,29,33]. Noteworthy, the precision of measurements obtained from the longitudinal plane is dependent on the appropriate cutting angle. Lastly, since the material is better condensed in a large cavity, the dimensions of the root-end cavity may also have an impact on the results’ accuracy. Retro-cavities are commonly standardized at a depth of 3 mm; however, their diameter is an underestimated and rarely specified parameter.

Generally, the better performance of Biodentine in terms of interfacial adaptability and sealing capacity is mostly associated with its improved formulation and physicochemical properties [9,10,29,37]. Despite the paucity of studies on the subject, Harvard MTA has also shown favorable physical properties [16,18,19,23]. The postulated paramount advantage of Biodentine, however, is its substantial biomineralization potential and associated interfacial interactions with dentine [38,39,40,41,42].

It is suggested that the ability of calcium silicates to induce hydroxyapatite deposition on the material surface leads to the formation of a Ca- and Si-rich intermediate interfacial layer with a tag-like structure, penetrating within the adjacent dentinal tubules. This phenomenon was mainly observed with Biodentine, ProRoot MTA, and certain other CSCs after material storage in phosphate buffered liquids (PBLs) for about 30 days [39,40,41], with one lone exception. A single study by Atmeh et al., reporting the development of an interfacial layer with Biodentine after only 4 days in distilled water. This layer, termed a “mineral infiltration zone,” was rich with carbonate ions [38]. The current evidence on this topic, however, is limited and yet equivocal in terms of the nature and quantitative characteristics of this intermediate layer and the relative significance of different CSCs in this regard [38,39,40,41]. Regretfully, a similar study on the biological activity of Harvard MTA is still lacking.

It is pertinent to highlight that none of the previously discussed SEM-based investigations has observed such an interfacial layer. Indeed, none of these studies utilized PBLs as immersion media; this is the most probable reason behind this. Ghorbanzadeh et al. examined the root-end marginal adaptability of three CSCs in phosphate-buffered saline over one week and two months, with no evidence of intermediate layer formation in any of the materials [43]. Further, the occurrence of this phenomenon has been reported mainly with the usage of simulated tissue fluids rather than with real ones such as blood or saliva.

Recently, a penetration or diffusion into dentinal tubules at a significant depth has been reported regarding certain CSCs, including Biodentine [41,44,45,46] as only one of these studies has addressed a retrograde application [46]. Commonly, all of them observed the phenomenon after sample storage at 100% humidity for only 2–4 weeks [41,44,46] or even in dry conditions [45]. The present findings are consistent with these investigations, as dentinal penetration was observed only with Harvard MTA (Figure 6). The short observation period may be the most reasonable explanation for this.

The three-dimensional seal of the resected apical root area is one of the most desired properties of root-end filling biomaterials, along with their biocompatibility and bioactivity. The marginal (interfacial) adaptability of CSCs is one of the main aspects of their capacity to hermetically seal the surgical site [6,7,24]. It is closely related to their ability to prevent the microleakage of fluids [31,36,47] and their dislodgement resistance [30]. Currently, there is no ISO standard that specifies the clinically acceptable limits for interfacial gap width in root-end filling materials. However, the marginal adaptation may have a considerable two-fold impact on the AMS healing process.

The material’s intra-tubular penetration ensures micro-mechanical adhesion to the underlying dentin, thus contributing to achieving a 3-dimensional physical seal of the apical area [41,45,46]. On the other hand, CSCs are capable of promoting biological seal through the apatite formation and inducing regeneration of the affected tissues [8,24,48]. In this context, the diffusion of the cement’s particles into the dentin canals can be considered as an initial phase of the biomineralization phenomenon. Our findings indicate that this diffusion becomes evident even during the material’s setting phase, as early as the first 48 h.

Essentially, good interfacial adaptation of CSCs is associated with their hydraulic nature and inherent properties such as wettability and external nano- and microporosity [8,23,49]. Further, the phenomenon of intra-tubular penetration itself might also be attributed to high-humidity conditions and a positive ratio between dentin canal diameter and material particle size [17]. They may facilitate the direct diffusion of the cement’s fine particles within dentinal canals even in the absence of PBL. Good handling characteristics also are a prerequisite for the material to adapt well to the dentin wall [28,46].

Reportedly, the material’s type also appeared as a determinative factor for marginal adaptability. MTA-like products originate from purified white Portland cement (PC) and share similarities in their chemical composition with PC [8,37,50,51,52]. Alite (tricalcium silicate) is the predominant crystalline phase in the vast majority of CSCs [37,52,53,54], including Biodentine as well [53,55]. Unlike conventional CSCs, Biodentine is developed synthetically via Active Biosilicate Technology. It is claimed to be composed mainly of highly purified TCS with small proportions of dicalcium silicate, calcium carbonate, and zirconium oxide as a radiopacifier [9,37,54]. Other authors suggested that the main phase in Biodentine is Hatrurite [56], which has been found in ProRoot MTA (90 wt%), MTA Angelus, and NeoMTA Plus as well [16,55].

The terms *Alite* and *Hatrurite* have been perceived as synonyms in PC chemistry [57], but in the case of CSCs, this would lead to a misapprehension of the matter [54,56]. Alite constitutes an impure form of tricalcium silicate (TCS) with varying phase content [51,52,57,58], while the genuine rock mineral Hatrurite has a steady composition, close to that of pure TCS. Synthetic Alite, however, is considered identical to Hatrurite [57,58].

On the other hand, Harvard MTA has a peculiar chemical content. According to Galal et al., it is composed of Merwinite (calcium magnesium orthosilicate) and Hatrurite, blended in a 75/20 wt% ratio, with Bi_2_O_3_ added as a radiopacifier [16]. Merwinite has not been detected in any other CSCs. Its favorable properties might be the primary rationale for the more favorable performance of Harvard MTA [16,59,60] in comparison with Biodentine, which is composed only of Hatrurite. Nevertheless, little is known regarding its impact as a Mg-bearing mineral substitute of Alite/Hatrurite on the properties of CSCs. Table 2 presents a comparison of the chemical composition and crystalline structure of typical industrial Alite, pure TCS, and the minerals Hatrurite and Merwinite [58,61,62,63].

In the last decade, ***calcium magnesium silicate*** ceramics, especially Merwinite-based ones, have attracted scientific interest. This class of biomaterials is suggested to possess great potential for medical applications due to their superior physical properties and bioactivity [59,60]. The formulations of Mg-containing CSs are based on the ternary system CaO–SiO_2_–MgO. In dentistry, however, this topic is still an under-studied area [64]. However, Harvard MTA is neither a typical cementitious material nor exactly an MS cement, as it is suggested to contain Merwinite instead of MgO [16,65]. Evidence on its hydration kinetics, and particularly those of Merwinite, is rather limited still [17,66,67].

Surprisingly, a recent study reported an entirely different chemical composition of Harvard MTA with no Mg in its formulation [17], being diametrically opposed to an earlier study and to the manufacturer’s information [16,68]. According to this work, it contains the typical calcium silicate phases, calcium aluminate, CaCO_3_, and CaWO_4_ (scheelite) as a radiopacifier [17]. These findings were not discussed by the authors in light of previous research, but this glaring contradiction emphasizes the limited knowledge on this novel biomaterial.

Conversely, the chemical composition of Biodentine seems to have been clarified. However, some controversies about its content still remain, especially with regard to the precise form of TCS used in its production and the presence of Mg or MgO in its formulation [37,52,56]. Obviously, “the apple of discord” is the inclusion of Mg/MgO in the cements’ formulae. Currently, fine distinctions between the materials come to the fore, since they may qualitatively impact the cement’s phase composition and affect its hydration process and properties [69,70,71].

Calcium silicate hydraulic cements have been defined as cutting-edge materials in the field of endodontics [24,72]. During the last three decades, they were extensively investigated, and a considerable amount of experimental and clinical evidence has been accumulated on both their properties and clinical performance. Despite the long application history of MTA, no ideal material has been developed to date. On the other hand, many new brands, defined as new generations, have been released in an attempt to overcome the disadvantages of the earlier products, many of them with modified or qualitatively different contents [69,70,73].

At this stage, some cognitive resonance begins to emerge as a natural consequence of this process. The old paradigm of “similarity,” which asserts that “there is no one superior ingredient that influences the ideal properties of these materials” [48,51], appears somewhat outdated. A new trend has been recently noted, aimed at more in-depth research in terms of ultrastructural analysis of chemical content and crystallinity and their impact on the hydration process and properties of hydraulic biomaterials [16,17,53,54]. To clarify the combined impact of major phases and minor constituents on the experimental and clinical performance of newly created CSCs, more interdisciplinary research is needed. The case of Harvard MTA exemplifies this, as it continues to harbor unresolved issues that require further investigation [16,17].

Finally, this study also had some **limitations:** (1). The usage of simulated tissue fluids was not included in the study design. (2). The SEM examination was performed just 48 h after material setting without any control groups for future monitoring. Further research is needed to evaluate the sealability of this material under experimental conditions more closely related to the real body environment to achieve more clinically relevant results. Among them, the short estimation period merits particular emphasis. CSCs come into contact with blood and saliva immediately after their application in the surgical site. Quite recent investigations suggested a volume and depth loss of incompletely set hydraulic calcium silicate cements used as root-end fillings in comparison with the completely set groups [74,75]. Therefore, the early setting period is of interest because it is crucial for the development of material properties and washout resistance.

## 5. Conclusions

Within the limitations of this in vitro study, Harvard MTA Universal demonstrated significantly better interfacial properties than Biodentine when applied as a root-end filling material. Moreover, an intra-tubular penetration of cement particles was observed but only with Harvard MTA. This novel calcium silicate brand could be regarded as a promising alternative for earlier CSCs in the context of apical microsurgery goals. However, further research is needed to clarify its proper content and hydration behavior, as well as their impact on its properties.

### Clinical Relevance

The quality of marginal adaptation of CSCs is a determinative feature for their clinical performance and the long-term prognosis of AMS. Therefore, it is essential for choosing the most suitable root-end filling material.

## Figures and Tables

**Figure 1 materials-18-04598-f001:**
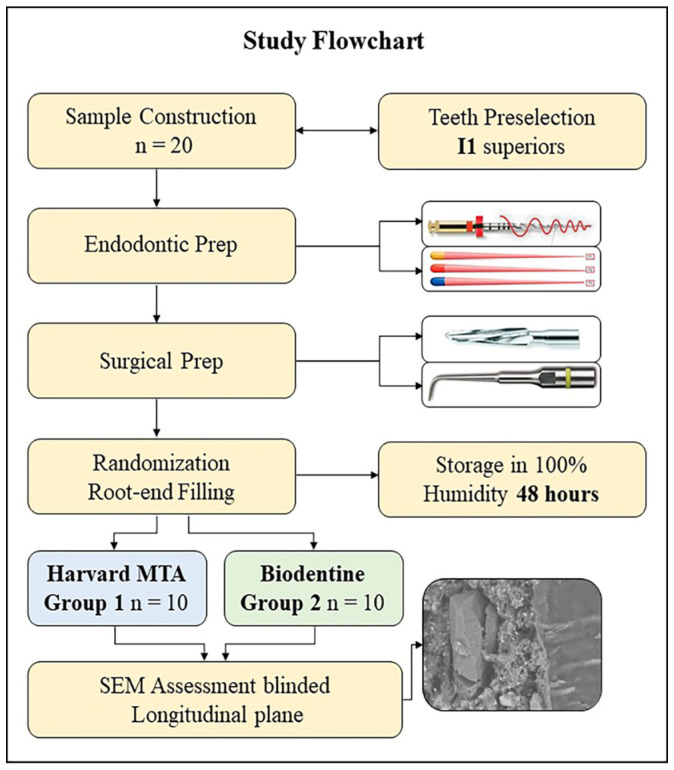
Flowchart of study protocol.

**Figure 2 materials-18-04598-f002:**
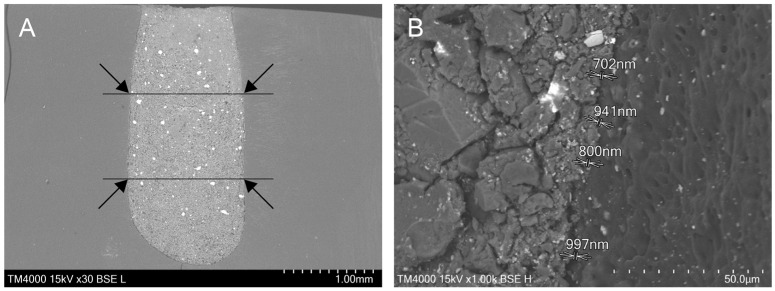
An example of the marginal gap quantification in SEM images: (**A**) four areas chosen initially (×30); (**B**) gap measurements at 4 points per micrograph (×1000).

**Figure 3 materials-18-04598-f003:**
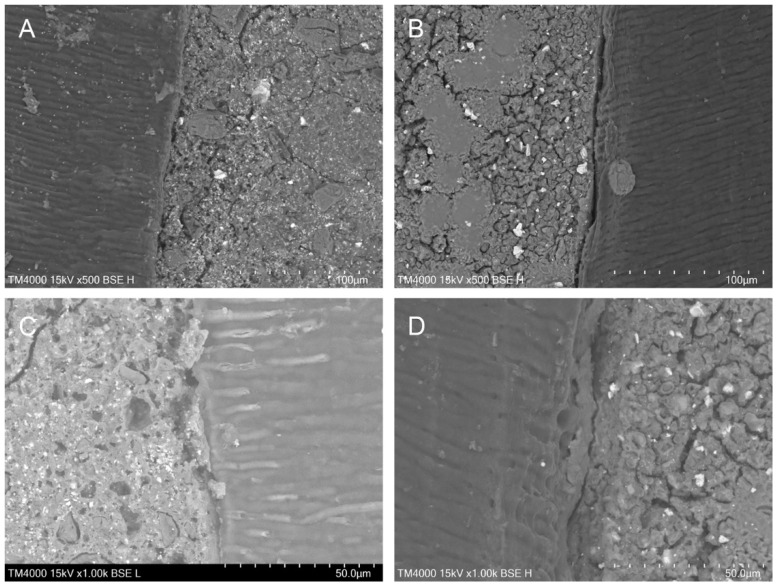
Representative SEM photomicrographs for the interfacial zone in the two groups at magnifications ×500 (first row) and ×1000 (second row): (**A**,**C**) Harvard MTA; (**B**,**D**) Biodentine; (**C**) penetration into the dentin tubules, observed in the Harvard MTA group.

**Figure 4 materials-18-04598-f004:**
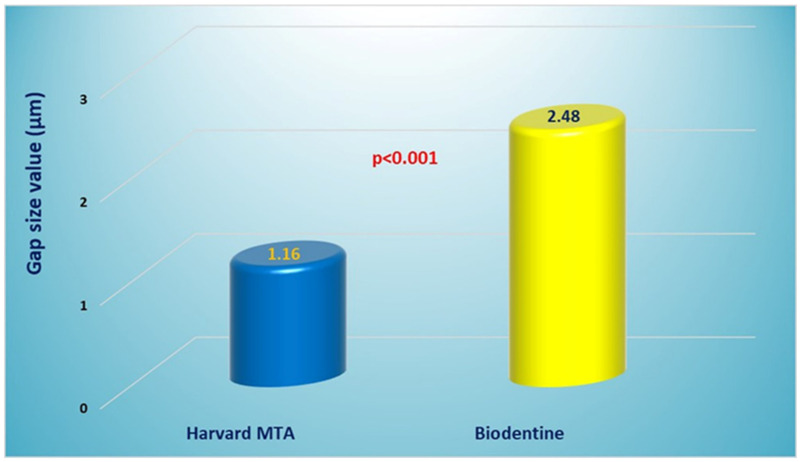
A bar chart presenting a comparison of the mean gap size values (in µm) found with Harvard MTA and Biodentine (*p* < 0.001; Cohen’s d = 3.52).

**Figure 5 materials-18-04598-f005:**
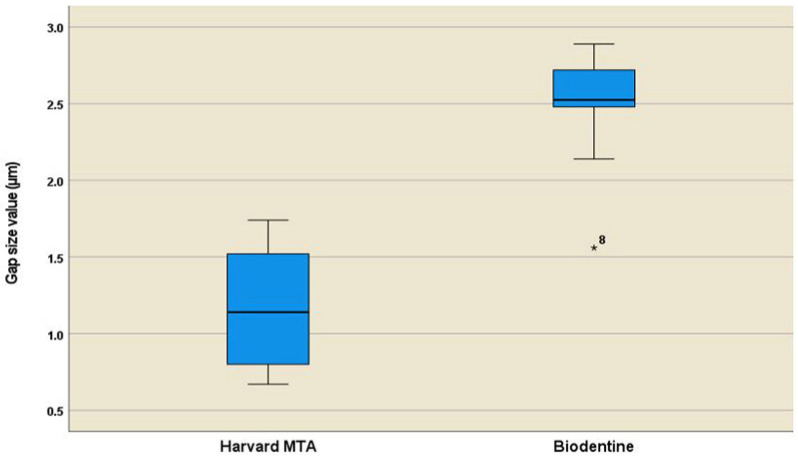
A boxplot diagram showing a graphical comparison of the sample medians and dataset’s characteristics exhibited in the Harvard MTA and Biodentine groups; whiskers indicate maximum and minimum values.

**Figure 6 materials-18-04598-f006:**
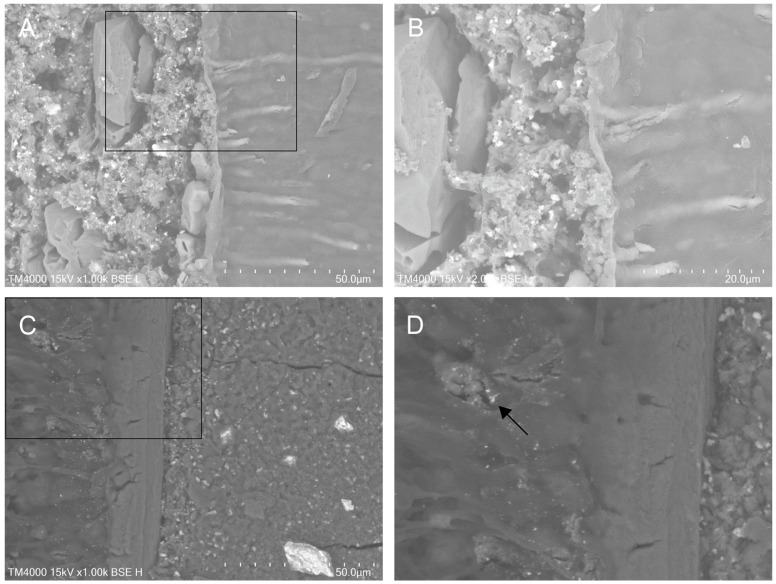
Illustrative images for Harvard MTA, showing its intimate proximity to the dentin wall with material penetration into the dentinal tubules: (**A**) ×1000 and (**B**) the framed area magnified (×2000); (**C**,**D**) an interfacial area revealing the presence of material particles within the open dentinal tubule in an overlying adjacent dentin wall (black arrow, ×1000); (**D**) the marked area magnified (×2000).

**Table 1 materials-18-04598-t001:** Comparative analysis of mean gap width (in μm), exhibited by Harvard MTA and Biodentine (descriptive statistics).

Material Type	N	Mean (µm)	Std. Error	95% CI for Mean	SD	Min	Max
Harvard MTA	10	1.16 **^a^**	0.12	0.89–1.42	0.37	0.67	1.74
Biodentine	10	2.48 **^b^**	0.12	2.20–2.75	0.38	1.56	2.89

Legend: CI—Confidence interval; SD—Standard deviation. Different superscript letters indicate statistically significant intergroup differences (*p* < 0.05).

**Table 2 materials-18-04598-t002:** Comparative chemical composition (in wt%) of Alite, TCS, Hatrurite, and Merwinite.

Mineral Content and Crystal Structure		Alite (Typical Composition)	Pure TCS (99.8% Purity)	Hatrurite	Merwinite
**Content** (wt%)	CaO	71.6%	73.6%	72.8%	49.96%
	SiO_2_	25.2%	26.2%	26.1%	35.5%
	Al_2_O_3_	1.0%	0.29%	0.4%	0.66%
	Fe_2_O_3_	0.7%	—	0.2%	—
	MgO	1.1%	0.1%	trace	11.62%
	Other oxides (Impurities)	P_2_O_5_-0.2%; Na_2_O-0.1%; K_2_O-0.1%, etc.	Free lime-0.31%	Ti, Al, P	FeO-1.22%
** *Crystal structure* **		*Polymorphic* structure *	*Triclinic*	*Trigonal*	*Monoclinic*

* A total of 7 polymorphs: Monoclinic (3), Triclinic (3), and 1 Rhombohedral with M1 and M3—most common.

## Data Availability

The raw data supporting the conclusions of this article will be made available by the authors on request.

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
