# Peer review of "Marginal Adaptability of Harvard MTA and Biodentine Used as Root-End Filling Material: A Comparative SEM Study"

_materials, 2025, doi:10.3390/ma18194598_

Round 1
Reviewer 1 Report
Comments and Suggestions for Authors
The manuscript "Marginal Adaptability of Harvard MTA and Biodentine Used as Endodontic Obturation Material: A Comparative SEM Study" addresses a clinically relevant topic in endodontic microsurgery and provides valuable comparative data on the interfacial adaptation of a new MTA formulation. However, several modifications are needed to strengthen the work, particularly in terms of textual conciseness, methodological details, and discussion of clinical implications.
Abstract:
It is important to briefly acknowledge the study's limitations, especially the short study period and in vitro design.
I suggest the authors simplify the methodological details and emphasize the study's clinical impact.
Introduction:
This section is extremely long and somewhat repetitive; the authors should adjust the introduction to condense the content into 4 to 5 paragraphs.
The authors should represent the research gap, reinforcing the limited evidence on the Harvard MTA.
I recommend concluding with clear objectives and a null hypothesis.
Materials and Methods:
The authors should provide a justification for calculating the sample size of n=10 per group, and specify whether it was based on previously published studies.
It would be beneficial to clarify the randomization and blinding procedure in more detail.
I recommend that the authors identify the software used for SEM measurements.
Was inter- or intra-rater reliability assessed?
A schematic diagram or flowchart of the experimental protocol would increase the clarity of the methodology.
Was the inclusion of an additional control material, e.g., ProRoot MTA, considered for a more robust comparison?
Results:
It would be beneficial to present exact p-values rather than just p < 0.05.
Adding boxplots to illustrate the distribution of data rather than just bar graphs would enrich the data.
SEM images can be standardized with consistent magnifications and reference markers.
I recommend that the clinical relevance of the tubular penetration observed in the Harvard MTA be added to this section.
Discussion:
This section should begin with a direct answer to the null hypothesis.
I recommend that the authors reduce redundancy; some sections related to chemical composition and crystallography are excessively detailed and distract from the clinical focus of the study.
Reinforcing the clinical implications of these findings, which would guide material selection in apical microsurgery, would benefit the discussion.
The authors should explicitly acknowledge the limitations, particularly the short evaluation time, lack of simulated tissue fluids, and small sample size.
Future perspectives for this research should be concisely included at the end of the discussion.
Conclusion:
The authors should rewrite this section in summary form, including two to three main findings.
Avoid definitive statements of superiority, given the study design, and emphasize that the Harvard MTA is promising but requires further validation.
Author Response
►Abstract:
►It is important to briefly acknowledge the study's limitations, especially the short study period and in vitro design.
Response: The requirement is fulfilled.
►I suggest the authors simplify the methodological details and emphasize the study's clinical impact.
Response: Fulfilled.
►Introduction: This section is extremely long and somewhat repetitive; the authors should adjust the introduction to condense the content into 4 to 5 paragraphs. The authors should represent the research gap, reinforcing the limited evidence on the Harvard MTA. I recommend concluding with clear objectives and a null hypothesis.
Response: Fulfilled.
►The authors should provide a justification for calculating the sample size of n=10 per group, and specify whether it was based on previously published studies. It would be beneficial to clarify the randomization and blinding procedure in more detail.
Response: Fulfilled.
►I recommend that the authors identify the software used for SEM measurements.
Response: Yes, an integrated program in the TM400 microscope software for measuring distances, lengths and angles was used for the quantification process, though a specific name or version of this program are not explicitly listed.
►Was inter- or intra-rater reliability assessed?
Response: Yes, details are given in the text.
►A schematic diagram or flowchart of the experimental protocol would increase the clarity of the methodology.
Response: Fulfilled.
►Was the inclusion of an additional control material, e.g., ProRoot MTA, considered for a more robust comparison?
Response: Yes, the inclusion of ProRoot MTA would contribute to the study's relevance and strength. It is the first developed MTA brand, representing the 1st generation of CSCs. Despite its proven good properties, it has shown some disadvantages and limitations. Its main physicochemical characteristics in comparison with representatives of the 2nd and 3rd generations were the subject of the PhD thesis of the first authors and have been presented in English in two book chapters.
►Results: It would be beneficial to present exact p-values rather than just p < 0.05.
Response: Fulfilled. In fact, the exact p-value (p < 0.001) was plotted on the background of Figure 3.
►Adding boxplots to illustrate the distribution of data rather than just bar graphs would enrich the data.
Response: Fulfilled.
►SEM images can be standardized with consistent magnifications and reference markers.
Response: Magnifications and reference markers are indicated in the lower bar of the images, except in Figure 6А. Unfortunately, due to an unintentional oversight in the screen settings, the design of this bar varies between transparent and black.
►I recommend that the clinical relevance of the tubular penetration observed in the Harvard MTA be added to this section.
Response: Special attention has been paid to these findings' clinical relevance in the Discussion section.
►Discussion: This section should begin with a direct answer to the null hypothesis.
Response: Fulfilled.
►I recommend that the authors reduce redundancy; some sections related to chemical composition and crystallography are excessively detailed and distract from the clinical focus of the study.
Response: Fulfilled.
►Reinforcing the clinical implications of these findings, which would guide material selection in apical microsurgery, would benefit the discussion.
Response: Fulfilled.
►The authors should explicitly acknowledge the limitations, particularly the short evaluation time, lack of simulated tissue fluids, and small sample size.
Response: Fulfilled.
►Future perspectives for this research should be concisely included at the end of the discussion.
Response: Fulfilled.
►Conclusion: The authors should rewrite this section in summary form, including two to three main findings.
Response: Namely, two to three main findings are indicated.
►Avoid definitive statements of superiority, given the study design, and emphasize that the Harvard MTA is promising but requires further validation.
Response: Fulfilled.

Reviewer 2 Report
Comments and Suggestions for Authors
General Assessment
The manuscript addresses an important topic in endodontics: the comparative marginal adaptability of two calcium-silicate cements (Harvard MTA Universal and Biodentine) used as root-end filling materials. The study is relevant for clinical decision-making in apical microsurgery and contributes new insights into the interfacial performance of Harvard MTA, a relatively new material. The text is generally well organized, but several aspects require clarification, refinement, and deeper discussion.
Major Comments
Novelty and Rationale
The study’s novelty lies in the evaluation of Harvard MTA compared with Biodentine, yet the introduction could better emphasize why this comparison is clinically significant given that Biodentine is considered the “gold standard.”
The rationale for expecting differences in marginal adaptability based on chemical composition should be explained more clearly.
Methodology
The description of randomization and blinding is not entirely clear. How was randomization performed, and how was operator blinding ensured during retrofilling?
The sectioning method (longitudinal only) may influence results. A justification of why this approach was chosen (and its implications for comparison with prior studies using transversal sections) is needed.
Please clarify whether the operator who measured the gaps in SEM images was calibrated, and if intra-examiner reliability was assessed.
Results
Figures should include scale bars and clearer labeling. Some micrographs are difficult to interpret without direct annotation.
Statistical results are valid, but reporting effect sizes (e.g., Cohen’s d) would strengthen the interpretation beyond p-values.
Discussion
The discussion is extensive but occasionally digresses into highly technical mineralogical details (e.g., polymorphism of Alite/Hatrurite). While interesting, some of this may not be essential for the dental audience. Consider condensing and focusing more directly on clinical implications.
The comparison with other studies is valuable, but inconsistencies in the literature could be synthesized in a more structured way (e.g., table summarizing previous findings).
The claim that Harvard MTA showed “significantly better adaptability” is correct for this dataset, but the conclusion should be more cautious given the small sample size (n=10 per group).
Limitations
The limitations are mentioned but understated. Key issues include the absence of simulated body fluids, short evaluation period (48h), and small sample size. These should be highlighted more prominently to avoid overinterpretation.
Minor Comments
Abstract: Consider stating explicitly the sample size per group in the Methods section.
Keywords: “magnesium silicate cements” may not be appropriate here, since the study did not directly investigate MSCs.
Figures: Ensure consistency in numbering (e.g., Figure 1 referenced in Methods vs. Results).
Language: The manuscript is generally clear, but some sentences are long and complex. Shortening them would improve readability.
References: Several references are older or non-peer-reviewed (e.g., company brochures). Ensure all claims, especially regarding chemical composition, are supported by robust scientific sources.
Recommendation
Major Revision
The study is interesting and potentially publishable, but the manuscript requires clarifications in methodology, refinement of the discussion, and a stronger emphasis on limitations. With these revisions, the contribution could be valuable to the field.
Author Response
Novelty and Rationale
►The study’s novelty lies in the evaluation of Harvard MTA compared with Biodentine, yet the introduction could better emphasize why this comparison is clinically significant given that Biodentine is considered the “gold standard.”
Response: Enough details are given regarding the two materials.
►The rationale for expecting differences in marginal adaptability based on chemical composition should be explained more clearly.
Response: More details are added.
Methodology
►The description of randomization and blinding is not entirely clear. How was randomization performed, and how was operator blinding ensured during retrofilling?
Response: Blinding was ensured at two levels: with respect to the SEM technician and statistician. Two sets of dental specimens, labeled "A" and "B," were given to the SEM technician in identical containers. The specimens were numbered 1 through 20 and lacked any other distinguishing marks. He was only told that the work involved quantifying dental material; no other information on its nature or trade names was provided. More details are given in the manuscript.
►The sectioning method (longitudinal only) may influence results. A justification of why this approach was chosen (and its implications for comparison with prior studies using transversal sections) is needed.
Response: Fulfilled.
►Please clarify whether the operator who measured the gaps in SEM images was calibrated, and if intra-examiner reliability was assessed.
Response: Fulfilled.
Results:
►Figures should include scale bars and clearer labeling. Some micrographs are difficult to interpret without direct annotation.
Response: Scale bars are indicated in the bottom bar of the images, except in Figure 6А. Unfortunately, due to an unintentional oversight in the screen settings, the design of this bar varies between transparent and black.
►Statistical results are valid, but reporting effect sizes (e.g., Cohen’s d) would strengthen the interpretation beyond p-values.
Response: Fulfilled.
Discussion:
►The discussion is extensive but occasionally digresses into highly technical mineralogical details (e.g., polymorphism of Alite/Hatrurite). While interesting, some of this may not be essential for the dental audience. Consider condensing and focusing more directly on clinical implications.
Response: Fulfilled.
►The comparison with other studies is valuable, but inconsistencies in the literature could be synthesized in a more structured way (e.g., table summarizing previous findings).
Response: The authors believe that the information has been analyzed and presented synthetically, rather than by sequentially listing the data of all authors cited. The present study is an original article, not a meta-analysis. Presenting such a table would be a good gift to those who intend to do so.
►The claim that Harvard MTA showed “significantly better adaptability” is correct for this dataset, but the conclusion should be more cautious given the small sample size (n=10 per group).
Response: This claim arises literally from the “significant difference” between materials under test as in “Conclusions” is stated that: “This novel biomaterial could be regarded as a promising alternative for…”.
Limitations
►The limitations are mentioned but understated. Key issues include the absence of simulated body fluids, short evaluation period (48h), and small sample size. These should be highlighted more prominently to avoid overinterpretation.
Response: The biomineralization potential is not the aim of the present study. Due to the many questions raised regarding the study design and its limitations, the authors provide a comparative table of experimental conditions adopted by similar studies.
Minor Comments:
►Abstract: Consider stating explicitly the sample size per group in the Methods section.
Response: The same answer as above.
►Keywords: “magnesium silicate cements” may not be appropriate here, since the study did not directly investigate MSCs.
Response: Fulfilled.
►Figures: Ensure consistency in numbering (e.g., Figure 1 referenced in Methods vs. Results).
Response: Fulfilled.
►Language: The manuscript is generally clear, but some sentences are long and complex. Shortening them would improve readability.
Response: Fulfilled. An attempt was made to improve it.
►References: Several references are older or non-peer-reviewed (e.g., company brochures). Ensure all claims, especially regarding chemical composition, are supported by robust scientific sources.
Response: Generally, older references were removed, except in cases with a scarcity of literature or when considering noteworthy or pioneer works with a great scientific impact (e.g., Han and Okuji, 2011).

Reviewer 3 Report
Comments and Suggestions for Authors
Dear Editor,
I have reviewed the manuscript entitled “Marginal Adaptability of Harvard MTA and Biodentine Used as Root-End Filling Material: A Comparative SEM Study” submitted for publication in Materials.
The study presents a comparative in vitro evaluation of Harvard MTA Universal and Biodentine as root-end filling materials, using SEM to assess marginal adaptability. It is a well-designed and timely study that provides valuable comparative data on the marginal adaptation of a novel calcium silicate cement (Harvard MTA) versus the established gold standard (Biodentine). The methodology is robust, employing blinding, randomization, and a rigorous SEM protocol for quantitative and qualitative analysis. The findings are noteworthy and of interest to the field. However, the manuscript requires significant revisions, primarily to the discussion section, before it is suitable for publication.
- The discussion section is overly long and includes speculative content, particularly regarding the chemical composition (e.g., Merwinite content) and hydration mechanisms of Harvard MTA. Since this study did not perform chemical analyses (such as XRD), these assumptions are not directly supported by the data and tend to distract from the main findings. The discussion should be shortened and more directly focused on interpreting the results superior adaptation and tubule penetration in relation to practical factors such as particle size, handling, or initial physical properties, rather than hypothetical hydration chemistry.
- The contradictory findings in the literature regarding Harvard MTA’s composition [13, 69, 70] should be highlighted more clearly and discussed as an unresolved uncertainty. Presenting this issue more critically will strengthen the transparency of the discussion and help readers understand the current gaps in knowledge.
- The evaluation period of 48 hours is relatively short and may not reflect the long-term performance of the materials. This should be stated more clearly as a limitation, since adaptation and stability can change over time.
- The absence of immersion in simulated body fluids (PBS, SBF) is a major methodological constraint. Because bioactivity and the formation of interfacial layers are both strongly time and environment dependent, this omission should be emphasized as a key factor limiting the clinical translatability of the findings.
- The definition of abbreviations should be added when first usage like "MTA"
- Affiliations ¹ and ² appear identical. Please correct.
- Ensure all items cited (Figure 4, Table 2) are present and correctly formatted.
- Some long sentences in the discussion could be simplified for better readability
Author Response
►1. The discussion section is overly long and includes speculative content, particularly regarding the chemical composition (e.g., Merwinite content) and hydration mechanisms of Harvard MTA. Since this study did not perform chemical analyses (such as XRD), these assumptions are not directly supported by the data and tend to distract from the main findings. The discussion should be shortened and more directly focused on interpreting the results superior adaptation and tubule penetration in relation to practical factors such as particle size, handling, or initial physical properties, rather than hypothetical hydration chemistry.
Response: The Discussion section was shortened and revised.
►2. The contradictory findings in the literature regarding Harvard MTA’s composition [13, 69, 70] should be highlighted more clearly and discussed as an unresolved uncertainty. Presenting this issue more critically will strengthen the transparency of the discussion and help readers understand the current gaps in knowledge.
Response: An effort has been made to meet this requirement An effort has been made to meet this requirement.
►3. The evaluation period of 48 hours is relatively short and may not reflect the long-term performance of the materials. This should be stated more clearly as a limitation, since adaptation and stability can change over time.
Response: This issue was analyzed in the Limitations section.
►4. The absence of immersion in simulated body fluids (PBS, SBF) is a major methodological constraint. Because bioactivity and the formation of interfacial layers are both strongly time and environment dependent, this omission should be emphasized as a key factor limiting the clinical translatability of the findings.
Response: The current study does not address the bioactivity of the materials under consideration. Similarly to other SEM investigations, it assesses only one of their physico-chemical properties.
- The definition of abbreviations should be added when first usage like "MTA"
- Affiliations ¹ and ² appear identical. Please correct.
- Ensure all items cited (Figure 4, Table 2) are present and correctly formatted.
- Some long sentences in the discussion could be simplified for better readability.
Response: Corrections have been made.

Round 2
Reviewer 1 Report
Comments and Suggestions for Authors
The revised version of the manuscript presents substantial improvements. The authors carefully addressed my recommendations, adding methodological details (randomization, blinding, reliability testing), improving the clarity of the figures and statistical reporting, and explicitly acknowledging the study's limitations. The discussion has been reorganized and now addresses the null hypothesis and highlights the clinical significance of the findings. The rewritten conclusion is concise and clearly acknowledges the findings.
I believe the manuscript is suitable for publication in Materials. I only recommend that the authors consider minor language polishing and further condensing the chemical composition section of the discussion, as it is still lengthy and repetitive.
I only recommend that the authors consider minor language polishing and further condensing the chemical composition section of the discussion, as it is still lengthy and repetitive.
Reviewer 2 Report
Comments and Suggestions for Authors
The manuscript has been revised and is now ready for publication.